# A Pilot Biomonitoring Study of Cumulative Phthalates Exposure among Vietnamese American Nail Salon Workers

**DOI:** 10.3390/ijerph17010325

**Published:** 2020-01-02

**Authors:** Julia R. Varshavsky, Rachel Morello-Frosch, Suhash Harwani, Martin Snider, Syrago-Styliani E. Petropoulou, June-Soo Park, Myrto Petreas, Peggy Reynolds, Tuan Nguyen, Thu Quach

**Affiliations:** 1School of Public Health, University of California, Berkeley, CA 94720, USA; rmf@berkeley.edu; 2Program on Reproductive Health and the Environment, University of California, Mailstop 0132, 550 16th Street, 7th Floor, San Francisco, CA 94143, USA; 3Department of Environmental Science, Policy and Management, University of California, Berkeley, CA 94720, USA; 4Environmental Chemistry Laboratory, California Department of Toxic Substances Control, Berkeley, CA 94720, USA; sharwani@hotmail.com (S.H.); martin.snider@outlook.com (M.S.); Sissy.Petropoulou@cdph.ca.gov (S.-S.E.P.); june-soo.park@dtsc.ca.gov (J.-S.P.); myrto.petreas@dtsc.ca.gov (M.P.); 5Cancer Prevention Institute of California, Berkeley, CA 94720, USA; peggy.reynolds@ucsf.edu (P.R.); tquach@ahschc.org (T.Q.); 6Department of Health Research and Policy, Stanford University School of Medicine, Palo Alto, CA 92705, USA; 7State Compensation Insurance Fund, Safety and Health Services, Santa Ana, CA 92705, USA; tuanscif@pacbell.net

**Keywords:** endocrine disrupting chemicals, exposure disparities, nail polish, occupational health, reproductive health, personal care products

## Abstract

Many California nail salon workers are low-income Vietnamese women of reproductive age who use nail products daily that contain androgen-disrupting phthalates, which may increase risk of male reproductive tract abnormalities during pregnancy. Yet, few studies have characterized phthalate exposures among this workforce. To characterize individual metabolites and cumulative phthalates exposure among a potentially vulnerable occupational group of nail salon workers, we collected 17 post-shift urine samples from Vietnamese workers at six San Francisco Bay Area nail salons in 2011, which were analyzed for four primary phthalate metabolites: mono-*n*-butyl-, mono-isobutyl-, mono(2-Ethylhexyl)-, and monoethyl phthalates (MnBP, MiBP, MEHP, and MEP, respectively; μg/L). Phthalate metabolite concentrations and a potency-weighted sum of parent compound daily intake (Σandrogen-disruptor, μg/kg/day) were compared to 203 Asian Americans from the 2011–2012 National Health and Nutritional Examination Survey (NHANES) using Student’s *t*-test and Wilcoxin signed rank test. Creatinine-corrected MnBP, MiBP, MEHP (μg/g), and cumulative phthalates exposure (Σandrogen-disruptor, μg/kg/day) levels were 2.9 (*p* < 0.0001), 1.6 (*p* = 0.015), 2.6 (*p* < 0.0001), and 2.0 (*p* < 0.0001) times higher, respectively, in our nail salon worker population compared to NHANES Asian Americans. Levels exceeded the NHANES 95th or 75th percentiles among some workers. This pilot study suggests that nail salon workers are disproportionately exposed to multiple phthalates, a finding that warrants further investigation to assess their potential health significance.

## 1. Introduction

Recent statements by the American College of Obstetricians and Gynecologists (ACOG) and other leading medical societies have called for timely action on reducing chemical exposures that contribute to developmental harm and reproductive health inequities among underserved populations [1,2]. As suggested in a commentary on the environmental injustice of the beauty care industry, eliminating harmful compounds in personal care products may provide opportunities for achieving this goal [3]. Indeed, many beauty industry workers are low wage immigrant women of reproductive age, a population highlighted by ACOG as uniquely vulnerable due to their disproportionate environmental exposures and social stressors at and outside of work [1].

The U.S. nail care industry has expanded rapidly over the last few decades, with active nail technician licenses totaling over 400,000 in 2015 [4] and recent nationwide spending up to $8.5 billion and $768 million on nail care services and nail polish alone [5]. Just over half of U.S. nail salon workers are Vietnamese, most of whom are women of reproductive age [4,6,7]. California has the largest number of manicurists and salons in the country [4,7], in part due to a rise in Vietnamese immigration between 1987 and 2002, which increased the proportion of Vietnamese nail salon workers from 10% to 60% [8,9].

Since many nail salon workers are low income women of color with limited English proficiency and access to chemical health and safety information, the California Healthy Nail Salon Collaborative and National Healthy Nail Salon Alliance formed in recent years to safeguard occupational health while maintaining economic integrity among this workforce [10,11,12,13]. Diverse coalition members include workers, owners, researchers, advocates, and government agency officials who contribute to local and national initiatives on chemical exposure research, worker health and safety trainings, green salon certification programs, and efforts to increase cosmetic product labeling requirements [13,14]. As part of this effort, California’s Department of Toxic Substances and Control (DTSC) revealed the presence of dibutyl phthalate (DBP), toluene, and formaldehyde in nail care products claiming to be free of “toxic trio” ingredients, and has since proposed nail salon products as a target category for alternatives assessment in the coming years [15,16,17]. Characterizing occupational exposures among this workforce can thus inform practical workplace intervention strategies and upstream efforts to promote safer products.

Phthalates are priority chemicals of concern because they are reproductive toxicants commonly used in nail care products and other cosmetics, for example, to increase flexibility and prevent nail polish from chipping or to sustain fragrances in scented lotions [17,18]. They are anti-androgenic compounds associated with multiple hormone mediated health impacts across the life course, including pregnancy complications, neurodevelopmental effects, cancer, and metabolic disease [19,20,21,22]. Moreover, in a groundbreaking 2008 report entitled, Phthalates and Cumulative Risk Assessment: The Tasks Ahead, the National Academy of Sciences (NAS) recommended evaluating phthalates in combination because their joint effects may increase risk of impaired male reproductive tract development during pregnancy [23]. 

Although two prior studies measured phthalate levels among U.S. nail salon workers, they were not designed to characterize exposures among Vietnamese or Asian workers specifically [24,25]. Study populations were either predominantly white (40% Asian) [25] or did not distinguish racial/ethnic differences (i.e., country of origin) among Asian study participants [24]. Additionally, worker levels were compared to the entire general population rather than a nationally representative sample of Asian Americans. Accordingly, the goals of this pilot biomonitoring study are to build on prior research and for the first time characterize phthalate exposures in an exclusive sample of Vietnamese nail salon workers compared to Asian Americans in the general population during an overlapping time period. Our study is also the first to characterize potency-weighted cumulative phthalates exposure (as recommended in the 2008 NAS report) among a uniquely exposed and potentially vulnerable workforce to reproductive toxicants [23].

## 2. Materials and Methods

### 2.1. Study Population and Sample Collection

Our study population comprised a subset of participants sampled in a 2011 pilot intervention study designed to educate nail salon workers and owners about how to reduce occupational exposures to various hazardous chemicals [26]. That study included 26 Vietnamese American workers at eight San Francisco Bay Area nail salons and measured each worker’s personal breathing zone air for volatile organic compounds such as toluene and methyl methacrylate. During air monitoring sessions, each participant completed questionnaires on characteristics such as age, the number of other workers and customers on the day of sampling, types of nail care services performed, and individual protective behaviors. Prior to educating workers about how to reduce their exposures, post-shift urine samples were collected from 17 of these workers at six nail salons during busier days of the week (Thursday, Friday, and Saturday) and warmer months of the year (June and July). Written consent was obtained in Vietnamese, and human subjects institutional review and approval from the Cancer Prevention Institute of California was received on 11 May 2011 (Protocol # 2010–013). Urine samples were collected in four-ounce glass containers, shipped on ice, and stored at −20 °C prior to analysis. To reduce potential effects of freeze-thaw cycles, samples were thawed and aliquoted (2–4 mL) in pre-cleaned amber vials. Water laboratory blanks were also prepared in the same type vials and frozen.

### 2.2. Laboratory Analysis

All urinary phthalate analyses were conducted at the Department of Toxic Substances Control (DTSC) in Berkeley, California. We measured four phthalate metabolites in this pilot study because our modest pilot study budget warranted a targeted approach to analyte selection, and our collaborators at DTSC sought to develop methods to measure urinary analytes for priority chemicals that would inform DTSC’s Safer Consumer Products program. We selected analytes based on earlier studies which suggested that mono-*n*-butyl-, mono-isobutyl-, mono(2-Ethylhexyl)-, and monoethyl phthalates (MnBP, MiBP, MEHP, and MEP, respectively) may be associated with working in a nail salon [24,25]. To analyze urine samples in the lab, we adapted and validated the Silva et al. (2004) method used by the Centers for Disease Control (CDC) to measure urinary MnBP, MiBP, MEHP, and MEP concentrations using liquid chromatography-tandem mass spectrometry (LC-MS/MS) [27]. Isotopically labeled internal standards (IS) were used for quantitation (MnBP IS was used for MiBP) and Agilent Nexus ABS Bond Elut cartridges for solid-phase extraction (SPE) clean-up. Samples were analyzed using a Varian 320MS LC-MS/MS with a Phenomenex Gemini C6-Phenyl column (100 mm × 2.00 mm, 3 µm). Baseline peak resolution was achieved for all metabolites, and no significant laboratory background was detected. Relative recoveries of quality control standards spiked into the urine before SPE ranged between 80% and 95%. Inter- and intra-day reproducibility and precision (coefficient of variation <15%) were demonstrated in addition to accuracy (<15% difference between expected and observed concentrations). The limit of detection (LOD) was defined as the lowest calibration standard providing a signal-to-noise ratio >6. Once the method was validated, all 17 nail salon worker urine samples were analyzed and aliquots were submitted to the clinical laboratory at San Francisco General Hospital for creatinine measurements.

### 2.3. Statistical Analysis

Nail salon worker data were compared to 2011–2012 CDC-administered National Health and Nutritional Examination Survey (NHANES), a nationally representative questionnaire and physical survey of the U.S. civilian, non-institutionalized population. As part of the NHANES medical examination, spot urine samples are collected and shipped on dry ice to CDC’s National Center for Environmental Health for analysis [28]. Demographic and laboratory files were downloaded from the NHANES website in May 2015 (http://www.cdc.gov/nchs/nhanes.htm). Out of 9756 NHANES participants, 2489 were sampled for >10 urinary phthalate metabolites, quantified using high performance liquid chromatography coupled with tandem mass spectrometry (HPLC-MS/MS) [29]. Primary comparison populations were restricted to pooled (men and women) and female-only NHANES Asian Americans 20–59 years old (*n* = 203 and *n* = 97, respectively). A secondary comparison population included all NHANES participants aged 20–59 years (*N* = 1175).

The nail salon urine sample and NHANES data were analyzed using SAS software, Version 9.4 (SAS Institute Inc., Cary, NC, USA). Phthalate metabolite concentrations below the LOD were imputed with LOD divided by √2. To correct for urine dilution, metabolite concentration (μg/L) was divided by urinary creatinine concentration (g/L), which is often used as a surrogate for urine dilution [30,31]. A potency weighted cumulative sum of phthalates daily intake (Σandrogen-disruptor, μg/kg/day) was constructed as described elsewhere [32,33]. Briefly, daily intakes of DnBP, DiBP, DEHP, and DEP were weighted by relative potency factors (RPF) and summed using the following equation: Σandrogen-disruptor = (DnBP × 1.0) + (DiBP × 0.24) + (DEHP × 0.61) + (DEP × 0.024). Daily intake for each phthalate was back-calculated from creatinine-corrected urinary metabolite concentrations (μg/g) using creatinine excretion rates and fractional excretion values of metabolites relative to parent compounds as follows: ((ME_i_ × CE)/(F_UE_ × 1000)) × (MW_p_/MW_m_), where i = individual phthalate, ME_i_ = creatinine-adjusted urinary metabolite concentration for each phthalate (μg/g), CE = body weight-adjusted creatinine excretion rate (mg/kg/day), F_UE_ = fractional molar ratio of excreted metabolite to ingested parent phthalate, MW_p_ = molecular weight of parent compound, and MW_m_ = molecular weight of metabolite. Since MEHP was the only DEHP metabolite included in this analysis, an *F_UE_* value of 0.062 was used for DEHP estimation, while and *F_UE_* value of 0.69 was used to estimate DnBP, DiBP, and DEP [34]. Average creatinine excretion rates of 23 mg/kg/day and 18 mg/kg/day were applied to men and women, respectively [35]. Concentrations *of* Σandrogen-disruptor and urinary metabolites were log-transformed prior to statistical testing to account for non-normal distributions, and statistical and marginal significance were defined at *p* < 0.05 and *p* < 0.10 for two-sided tests, respectively.

Descriptive statistics were calculated for worker and salon characteristics, such as average hours worked per week, number of services performed on the day of sampling, and salon volume in cubic meters of air (m^3^). We calculated detection frequencies (% > LOD) of measured urinary metabolites and performed univariate statistics on creatinine-corrected concentrations, non-creatinine-corrected concentrations, and our cumulative exposure metric (Σandrogen-disruptor). Univariate statistics included the arithmetic mean, geometric mean (GM), geometric standard deviation (GSD), or geometric standard error (GSE), range, and 25th, 50th, 75th, and 95th percentiles, and were calculated for all 17 nail salon workers and female-only workers (*n* = 15). These were also calculated for NHANES comparison populations, accounting for sample population weights and the complex NHANES survey design according to analytical guidelines [28].

Phthalate concentrations were compared between nail salon workers and NHANES 2011–12 participants using a parametric *t*-test for MnBP, MEHP, MEP, and the cumulative exposure metric (Σandrogen-disruptor). For statistical hypothesis testing, NHANES values were assumed to be the true population values. The Wilcoxon signed rank non-parametric *t*-test was used for the MiBP comparison, since several MiBP outliers were observed among nail salon workers. This analysis was performed separately for pooled (men and women) and female-only Asian Americans aged 20–59 years old, in addition to all race/ethnicities of the same age range.

Since urine dilution correction and daily intake estimation may both introduce uncertainty and variability into back-calculated biological exposure metrics [30,31,36,37], we conducted a supplementary analysis to evaluate whether alternate approaches impacted results. Two additional cumulative exposure metrics were calculated by first applying RPF weights directly to non-creatinine-corrected concentrations measured in urine (μg/L), and then by applying RPF weights to creatinine-corrected urinary metabolite concentrations (μg/g) [32]. Compared to the original metric (Σandrogen-disruptor), which both corrected for urine dilution and estimated daily intake, the first supplementary metric provided a comparison to an absence of both potential sources of uncertainty/variability, while the second provided a comparison to the absence of daily intake estimation only.

To facilitate interpretation of results, we additionally compiled a summary of urinary phthalate metabolite concentrations measured in relevant biomonitoring and health impact studies. These included two previous biomonitoring studies that reported creatinine-corrected values, one that collected 25 post-shift urine samples from predominantly Asian nail salon workers in Maryland, USA [24], and another that collected 30 urine samples exclusively from the general population in Vietnam [38]. One other known study measured urinary MnBP and MiBP in U.S. nail salon workers, but the study population was predominantly white and specific gravity-corrected concentrations were the only values reported, which limited comparability with our results [25]. Lastly, we included five epidemiology studies that reported significant associations between non-corrected concentrations of urinary metabolites measured in this study and adverse male developmental outcomes [39,40,41,42,43], excluding studies that reported specific-gravity corrected values only, as they were not directly comparable.

## 3. Results

All 17 nail salon workers were born in Vietnam and most preferred to speak Vietnamese at home (Table 1). The average age of the study population was 40 years. There was a wide range of hours worked per week at the salon (10–60 h), with a mean of 34.5 h per week, and number of services performed on the day of sampling (1–17 services), with an average of eight services. The average number of other workers in the salon on the day of sampling was four and ranged from two to eight. The estimated number of total salon customers on the day of sampling was around 22, with a range between seven and 40. Most workers reported using gloves and a metal trash bin. The most popular ventilation practice was leaving the doors of the salon open.

Detection frequencies for urinary phthalate metabolites were similar or higher in nail salon workers (93%–100%) compared to NHANES Asian Americans (80%–100%; Table 2 and Table 3). In all workers (*n* = 17), MEP had the highest creatinine-corrected GM of 38 μg/g (GSD = 3.3, range = 8.1–497 μg/g), followed by 23 μg/g for MnBP (1.7, 9.9–61 μg/g), 13 for MiBP (2.3, 3.6–121 μg/g), and 5.9 for MEHP (1.8, 2.2–16 μg/g; Table 2). The same rank order was observed among female-only workers and NHANES Asian Americans (pooled and female participants; Table 2 and Table 3), as well as for non-creatinine-corrected values (Table 4).

Creatinine-corrected phthalate metabolite concentrations were higher among nail salon workers compared to NHANES Asian Americans, and some of the differences were statistically significant (Table 2). MnBP, MiBP, MEHP, and MEP were 186% (*p* < 0.0001), 66% (*p* = 0.015), 158% (*p* < 0.0001), and 25% (*p* = 0.445) higher, respectively. For some nail salon workers, creatinine-corrected metabolite concentrations exceeded the NHANES 95th percentile in adult Asian Americans. The highest worker concentrations were 44%, 293%, 20%, and 40% above the NHANES 95th percentile for MnBP, MiBP, MEHP, and MEP, respectively. Additionally, two nail salon workers had more than one urinary metabolite concentration greater than the NHANES 95th percentile (Table 2). Female nail salon workers had similar but attenuated results, with 132% (*p* < 0.0001), 46% (*p* = 0.041), 120% (*p* = 0.0002), and 27% (*p* = 0.455) higher MnBP, MiBP, MEHP, and MEP concentrations, respectively, compared to NHANES Asian American females (Table 3).

Non-creatinine-corrected results for pooled (men and women) and female participants were also similar and/or slightly attenuated, though MiBP was not significantly higher in pooled nail care workers compared to NHANES Asian Americans, and only marginally significant when the analysis was restricted to female participants (Table 4). Because urinary concentrations among all NHANES race/ethnicities (*N* = 1175) were lower than those measured in NHANES Asian Americans for each phthalate metabolite except MEP, this study population was not included in further analyses.

Among pooled study populations (men and women), the GM of cumulative phthalates daily intake (Σandrogen-disruptor) was 97% higher in nail salon workers (GM = 2.6; GSD = 1.7 µg/kg/day) compared to NHANES (GM = 1.3; GSE = 1.1 µg/kg/day; *p* < 0.0001; Figure 1).

Among women, the percent difference between workers and NHANES Asian Americans was 81% (*p* = 0.0004). However, the maximum NHANES cumulative exposure of 39 µg/kg/day was over five times greater than the maximum nail salon worker cumulative exposure of 6.7 µg/kg/day. The Σandrogen-disruptor distributions for pooled and female-only NHANES Asian Americans were very similar, except for at the upper tail end of the distribution, where most extreme cumulative exposure concentrations belonged to women. Thus, the NHANES 95th percentile for women (8.5 µg/kg/day) was substantially higher than the pooled NHANES 95th percentile (5.2 µg/kg/day). None of the nail salon workers exceeded the female-only NHANES 95th percentile, but two nail salon workers (both women) exceeded the pooled NHANES 95th percentile. Twelve workers (70%), including both men, were above the NHANES 75th percentile for female-only and pooled Asian Americans (Figure 1).

The supplementary analysis exploring alternate cumulative exposure metrics (i.e., alternate approaches to urine dilution correction and daily intake estimation) produced similar results between supplementary cumulative exposure metrics and our original cumulative exposure metric (Σandrogen-disruptor). This indicates that differences in cumulative phthalates exposure between nail salon workers and the Asian American general population did not depend on whether or not urine dilution corrections or parent compound back-calculations were applied to phthalate metabolite concentrations measured in urine. We thus reported results solely for the original cumulative phthalates exposure metric (Σandrogen-disruptor).

All creatinine-corrected urinary phthalate metabolite concentrations measured among nail salon workers in this study were similar to or higher than concentrations reported in a previous biomonitoring study of the general population in Vietnam (Table 5) [38]. However, lower creatinine-corrected urinary concentrations were observed in this study compared to nail salon workers sampled in 2003–2004, except for MiBP which was twice as high in our nail salon worker population [24]. Besides MEP, non-creatinine corrected metabolite concentrations in this study were similar to or higher than those measured in epidemiology studies that reported a significant association with adverse male developmental endpoints, such as decreased anogenital distance (AGD), reductions in penile length, and reduced masculine play among boys (Table 6) [39,40,41,42].

## 4. Discussion

This pilot study of 17 nail salon workers provides evidence of elevated individual metabolite and cumulative phthalates exposure in comparison to a national sample of the U.S. Asian American population. Specifically, nail salon workers had significantly higher levels of MnBP, MiBP, and MEHP, as well as higher levels of exposure to multiple phthalates that were combined in a summary metric based on their relative toxicities (Σandrogen-disruptor). To our knowledge, this is the first biomonitoring assessment of phthalates in an exclusive sample of Vietnamese nail salon workers in California, who comprise a uniquely vulnerable occupational group. This pilot study suggests that nail salon workers are exposed to higher levels of biologically relevant phthalate combinations than the general population.

Individual metabolite concentrations exceeded the NHANES 95th percentile in several nail salon workers, and for more than one metabolite in two participants, indicating that some workers may be simultaneously exposed to phthalate combinations at very high levels relative to the general Asian American population. Most nail salon workers had higher cumulative phthalates daily intake than the NHANES 75th but not 95th percentiles. However, the NHANES Asian American comparison population conceivably might include Vietnamese nail salon workers, which could have skewed the distribution (and 95th percentile) towards the upper end of extreme values. Due to existing evidence that combined phthalate exposures may have greater adverse health effects than individual phthalate exposures, future studies in a larger sample of nail salon workers could examine whether these pilot findings can be replicated [23,44,45,46].

While this analysis did not assess direct connections between phthalates and health impacts, MnBP, MiBP, and MEHP concentrations were similar to or above those measured in studies reporting adverse developmental endpoints in male offspring [39,40,41,42,44]. For example, in a large prospective cohort study across four major U.S. cities (*N* = 753) that found a significant association between MEHP and reduced anogenital distance among baby boys, Swan et al. (2015) reported a non-creatinine corrected GM of 1.9 ng/mL (95% CI: 1.8, 2.1), which was >1.5 times lower than the GM of 3.5 ng/mL (95% CI: 2.4, 5.3) observed among female nail salon workers in this study [40]. These exposure levels are of potential concern for women in the nail salon workforce who are of reproductive age and may become pregnant.

MnBP, MEHP, and MEP concentrations measured in this study were lower than those collected seven years prior, reflecting U.S. biomonitoring trends for these metabolites between 2001 and 2012 [24]. Similarly, MiBP concentrations were two times higher in the current nail salon worker population, which is consistent with a reported >100% increase in urinary MiBP levels over the last decade [47]. These temporal changes may reflect product reformulations in response to increased phthalates scrutiny and consumer preferences for safer alternatives over the last decade [47,48,49]. Due to the U.S. Food and Drug Administration’s (FDA) limited oversight over cosmetic ingredients, large public campaigns in recent years have pressured beauty companies, such as OPI, Sally Hansen, and Revlon, to remove phthalates from nail polish and other personal care products [50]. Subsequent data suggest that DiBP may be replacing the better-known DnBP, which has long been used as a hardener in nail polish to preserve shape and color [47,48,51,52,53]. However, DnBP is still detected in many nail care products, including those claiming to be phthalate free [15,17], and both compounds contribute to androgen-mediated risks associated with phthalate co-exposures [23,54]. Continued product testing and biomonitoring research will be required to evaluate these trends going forward.

While absolute levels have changed over time, the difference between nail salon workers and the general population remained relatively constant between the previous study (which compared samples collected from workers in 2003–2004 to NHANES 2001–2002) [24] and our study (which compared samples collected from workers in 2011 to NHANES 2011–2012). Both studies found higher MnBP and MiBP concentrations among nail salon workers, which is consistent with common use of their parent compounds in nail and other beauty products [25,47]. MEP levels were substantially but not significantly higher among nail salon workers in both studies. Although typically associated with personal care product use in non-occupational populations [18,19,25,55], MEP exposures at the nail salon may be negligible relative to high background concentrations measured in the general population. Indeed, MEP levels were the highest observed in our study as well as previous studies assessing this compound in the U.S. population [32,47].

Our MEHP results are also consistent with prior research, which found the highest MEHP concentrations among nail salon workers [24], despite DEHP’s prevalent association with dietary intake from food packaging and processing equipment, rather than from cosmetic or beauty product use [44]. Levels were nearly five times that of U.S. adults and higher than every other occupational group evaluated (including phthalate and PVC manufacturing workers) [24]. Oxidative (secondary) metabolites of DEHP were not significantly elevated in nail salon workers, which suggested the possibility of an unknown DEHP source that may be unique to nail salons. We speculate that acrylic nail powder may be a possible unique MEHP exposure source in this workforce, as it contains a catalyst that metabolizes to benzoic acid [56,57]. Given that MEHP is an ester derivative of benzoic acid, acrylic nail powder may conceivably be a direct source of MEHP exposure in nail salons, which would explain why higher primary but not secondary metabolites (which can only be formed in the liver) were observed previously. However, a major limitation of our study is that we did not measure oxidative metabolites for DEHP, which constitute a larger fraction of absorbed parent compound dose than MEHP. Consequently, measuring MEHP alone might underestimate DEHP daily intake. Additionally, oxidative metabolites are less sensitive than primary monoesters to laboratory contamination because they are formed solely from phthalate metabolism in the liver [58]. Future biomonitoring studies that evaluate DEHP should include at least one oxidized metabolite.

Other study limitations should be addressed in future work. First, our sample size was too small to predict phthalate metabolite and cumulative exposure concentrations from questionnaire data on worker and salon characteristics, thus limiting statistical power to identify specific occupational sources of phthalates exposure. Forthcoming research on nail salon exposures should also include the collection of pre-shift urine samples in addition to air samples collected during the shift, which would help differentiate between occupational and non-occupational (environmental) exposures. The addition of pre-shift urine samples may be the best way to assess phthalate exposures in nail salon workers, given the short metabolic half-lives of phthalates (less than 24 h) [26]. Including questionnaire data about non-occupational exposure sources, such as at-home personal care product use and dietary behaviors, would also facilitate delineation between occupational and environmental exposures. Additionally, while prior research has demonstrated moderate long-term reliability of urine spot samples in adults for some metabolites, including MEP and MnBP, studies evaluating reproducibility in other populations are somewhat equivocal [59,60,61]. Furthermore, intra-person variability may be higher in the workplace due to short-term exposure spikes from specific job tasks [26]. Longitudinal data collection could help resolve this issue in future studies.

Lastly, although nail salon workers in this study were compared to NHANES Asian Americans of the same age range sampled within a year of workers in this study, some non-occupational differences between the groups could potentially explain exposure disparities. For example, workers in this study were Vietnamese-born, while the NHANES Asian American population may include second or third generation immigrants. Vietnamese immigrants might use different personal care products or eat different foods than those born in the United States, which could result in observed exposure disparities between workers and the NHANES sample that are not occupationally based. However, Guo et al. (2011) biomonitored 30 people in Vietnam and reported equal or lower creatinine-corrected phthalate metabolite levels than workers in this study [38], suggesting our findings are not likely to due to cultural differences in personal care product use or diet originating from the country of origin. Other differences between the groups, such as location, sampling strategy (convenience rather than multistage probability), data collection time of day (post-shift compared to random), and month or season of sampling (summer rather than throughout the year), might also partially explain exposure differences.

## 5. Conclusions

Despite limitations, findings from this pilot study suggest that nail salon workers are disproportionately exposed to multiple phthalates daily at levels associated with adverse effects on reproductive health and development. Comparing Vietnamese nail salon workers to a national sample of Asian Americans provides useful insight and supportive evidence that higher concentrations of individual metabolites and potency weighted cumulative phthalates levels may be due to workplace exposures. Further research should more broadly characterize phthalate exposures and their health implications in this workforce. Continued efforts should be made to identify workplace practices that minimize exposure (i.e., ventilation and glove use) as well as upstream opportunities for promoting safer alternatives.

## Figures and Tables

**Figure 1 ijerph-17-00325-f001:**
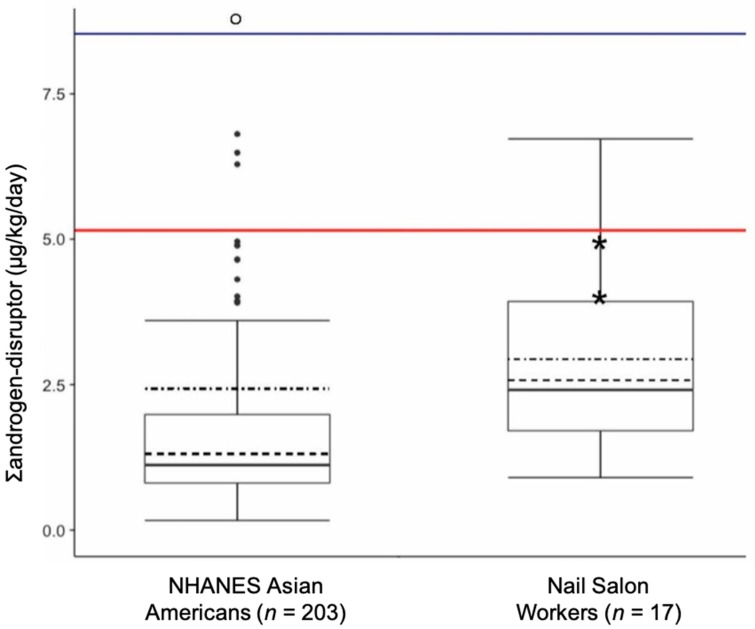
Comparison of cumulative phthalates daily intake (Σandrogen-disruptor) between nail salon workers and NHANES 2011–2012 Asian Americans. Boxes represent interquartile range (IQR: 25th–75th percentiles). Dark lines represent medians. Dashed lines represent geometric means. Dot-dashed lines represent arithmetic means. Whiskers extend to min and max (Max = most extreme values within 1.5 • IQR of the median for NHANES). NHANES outliers are represented by dark points, and hollow point denotes outliers off the *y*-axis scale (a total of seven, five of which were women). Red line represents 95th percentile for pooled NHANES Asian Americans (*n* = 203). Blue line represents 95th percentile for female NHANES Asian Americans (*n* = 97). *p*-value of statistical tests of difference between nail salon workers and NHANES Asian Americans <0.0001. ***** Indicates male nail salon worker observations (*n* = 2).

**Table 1 ijerph-17-00325-t001:** Characteristics of Vietnamese nail salon workers in California (*n* = 17).

Worker and Salon Characteristics	Mean (Range) * or *n* (%)
Age	40 (23–57) *
Sex	
*Female*	15 (88%)
*Male*	2 (12%)
Birthplace	
*Vietnam*	17 (100%)
Preferred language(s)	
*Vietnamese*	17 (100%)
Average work hours per week *^a^*	34.5 (10–60) *
Salon volume (m^3^)	274 (42–437) *
Number of services performed that day *^a^*	8 (1–17) *
Number of other workers that day *^a^*	4 (2–8) *
Number of salon customers that day *^a^*	22 (7–40) *
General reported glove use *^a^*	10 (67%)
Metal trash bin with tight fitting lid use *^a^*	9 (60%)
Ventilation practices that day *^a^*	
*Used a table fan*	7 (47%)
*Left windows open*	5 (33%)
*Left doors open*	14 (93%)
*Other ventilation practices*	10 (59%)
*Total number of ventilation practices*	3 (0–5) *

*^a^* Restricted to *n* = 15 due to missing survey data for two workers. * Indicates range rather than *n* (%).

**Table 2 ijerph-17-00325-t002:** Comparison of creatinine-corrected phthalate metabolite concentrations (μg/g) between all nail salon workers and 2011–2012 NHANES Asian Americans *^a^*.

Phthalate Metabolites	All Nail Salon Workers (*n* = 17)				All NHANES Asian Americans (*n* = 203)			
	GM (GSD)	Range	% >LOD *^b^*	% >NH_p95_	GM (GSE)	Range	% >LOD *^b^*	NH_p95_
Mono-*n*-butyl phthalate (MnBP)	23 (1.7)	9.9–61	94%	18%	8.2 (1.1)	0.44–930	90%	42
Mono-isobutyl phthalate (MiBP)	13 (2.3)	3.6–121	100%	12%	7.6 (1.1)	0.36–90	98%	31
Mono(2-Ethylhexyl) phthalate (MEHP)	5.9 (1.8)	2.2–16	94%	6%	2.3 (1.1)	0.18–155	80%	14
Monoethyl phthalate (MEP)	38 (3.3)	8.1–497	100%	6%	301 (1.1)	0.75–1876	100%	355

NH_p95_ = NHANES 95th percentile. LOD = Limit of detection. *^a^ p*-values of statistical tests of difference between all nail salon workers and NHANES Asian Americans: MnBP (<0.0001), MiBP (0.015), MEHP (<0.0001), and MEP (0.445). *^b^* Nail salon worker LOD = 1.20 μg/L for MnBP, MiBP, and MEHP, and 3.56 μg/L for MEP. NHANES LOD = 0.40, 0.20, 0.50, and 0.60 μg/L for MnBP, MiBP, MEHP, and MEP, respectively.

**Table 3 ijerph-17-00325-t003:** Comparison of creatinine-corrected phthalate metabolite concentrations (μg/g) between female nail salon workers and 2011–12 NHANES Asian American females *^a^.*

Phthalate Metabolites	Female Nail Salon Workers (*n* = 15)				Female NHANES Asian Americans (*n* = 97)			
	GM (GSD)	Range	% >LOD *^b^*	% >NH_p95_	GM (GSE)	Range	% >LOD *^b^*	NH_p95_
Mono-*n*-butyl phthalate (MnBP)	22 (1.6)	9.9–55	93%	13%	9.3 (1.1)	0.28–371	90%	44
Mono-isobutyl phthalate (MiBP)	11 (1.9)	3.6–62	100%	7%	7.5 (1.1)	0.14–66	98%	28
Mono(2-Ethylhexyl) phthalate (MEHP)	5.6 (1.9)	2.2–16	93%	7%	2.6 (1.2)	0.35–160	80%	15
Monoethyl phthalate (MEP)	44 (3.3)	8.1–497	100%	7%	34 (1.1)	0.42–1293	100%	298

NH_p95_ = NHANES 95th percentile. LOD = Limit of detection. *^a^ p*-values of statistical tests of difference between female nail salon workers and NHANES Asian American females: MnBP (<0.0001), MiBP (0.041), MEHP (0.0002), and MEP (0.455). *^b^* Nail salon worker LOD = 1.20 μg/L for MnBP, MiBP, and MEHP, and 3.56 μg/L for MEP. NHANES LOD = 0.40, 0.20, 0.50, and 0.60 μg/L for MnBP, MiBP, MEHP, and MEP, respectively.

**Table 4 ijerph-17-00325-t004:** Comparison of non-creatinine-corrected phthalate metabolite concentrations (μg/L) between nail salon workers and 2011–2012 NHANES Asian Americans *^a^.*

Phthalate Metabolites	All Nail Salon Workers (*n* = 17)	All NHANES Asian Americans (*n* = 203)		Female Nail Salon Workers (*n* = 15)	Female NHANES Asian Americans (*n* = 97)		Male Nail Salon Workers (*n* = 2)
	GM (GSD)	GM (GSE)	*p*-Value *^b^*	GM (GSD)	GM (GSE)	*p*-Value *^b^*	Observed Values
MnBP	16 (2.4)	6.1 (1.2)	0.007	14 (2.1)	5.6 (1.2)	0.014	20, 105
MiBP	8.5 (3.0)	5.6 (1.1)	0.147	7.0 (2.2)	4.4 (1.1)	0.056	7.1, 209
MEHP	3.9 (2.2)	1.7 (1.1)	<0.001	3.5 (2.0)	1.6 (1.1)	0.001	5.9, 16
MEP	26 (3.5)	23 (1.1)	0.681	27 (3.7)	21 (1.1)	0.425	15, 16

*^a^* Nail salon worker limit of detection (LOD) = 1.20 µg/L for MnBP, MiBP, and MEHP, and 3.56 µg/L for MEP. NHANES LOD = 0.40, 0.20, 0.50, and 0.60 μg/L for MnBP, MiBP, MEHP, and MEP, respectively. *^b^*
*p*-values from statistical tests of difference between nail salon workers and NHANES.

**Table 5 ijerph-17-00325-t005:** Comparison of creatinine-corrected GM (range) of phthalate metabolite concentrations (μg/g) measured in this study to previous biomonitoring studies.

Metabolites	This Study	Hines et al. (2009) [24]	Guo et al. (2011) [38]
MnBP	23 (3.0–105)	34 (<LOD–199)	17 (8.0–47)
MiBP	13 (1.7–209)	6.3 (<LOD–27)	12 (5.7–34)
MEHP	5.9 (<LOD–16)	19 (<LOD–1480)	2.4 (<LOD–5.3)
MEP	38 (5.7–399)	119 (17–1580)	6.5 (1.8–42)
**Study details**	17 post-shift samples collected from 100% Vietnamese nail salon workers in California, USA, 2011	25 post-shift urine samples collected from 88% Asian nail salon workers in Maryland, USA, 2003–2004	30 samples collected from general population in Vietnam, 2006–2007

**Table 6 ijerph-17-00325-t006:** Comparison of non-creatinine-corrected phthalate metabolite concentrations (μg/L) measured among female nail salon workers in this study to previous epidemiology studies that reported significant associations between prenatal exposures and male developmental endpoints *^a^.*

Metabolites	This Study					Swan et al. (2005) [44]	Swan et al. (2010) [39]	Suzuki et al. (2012) [42]	Bustamante-Montes et al. (2013) [41]	Swan et al. (2015) [40]
	GM	Mean	Range	Median	p75	Median (p75)			Mean (Range)	GM (p75)
MnBP	14	17	3.0–38	13	28	14 (31) **	13 (28) **	47 (65)	0.65 (0.25–1.6)	6.4 (17)
MiBP	7.0	9.1	1.7–26	8.5	13	2.5 (5.1) **	2.4 (5.1) **	n/a	n/a	4.0 (11)
MEHP	3.5	4.5	0.9–13	3.2	6.9	3.3 (9.0)	2.9 (6.2)	3.7 (7.1) **	4.0 (0.4–20) **	1.9 (4.7) **
MEP	27	66	4.7–399	18	68	128 (437) **	n/a	7.8 (32)	7.6 (0.27–27)	28 (81)
Sample size (N)	17					85	74	111	73	753

p75 = 75th percentile. Green color indicates associated metabolite concentrations that were similar or below those measured in this study. *^a^* Developmental endpoints examined in each study: Reduced anogenital distance (Swan et al. (2005, 2015) [40,44], Suzuki et al. (2012) [42], and Bustamante-Montes et al. (2013) [41]], reduced penile size [Bustamante-Montes et al. (2013) [41]], and reduced masculine play in young boys [Swan et al. (2010) [39]). ** Denotes significant association.

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
