# Peer review of "A Pilot Biomonitoring Study of Cumulative Phthalates Exposure among Vietnamese American Nail Salon Workers"

_ijerph, 2020, doi:10.3390/ijerph17010325_

Round 1

Reviewer 1 Report

This is a very solid study, well written and with a well thought out analysis.  If indeed there are only two other published research studies on phthalate exposures among nail salon workers in the US, it is a solid contribution to a much needed literature.  The authors explained their methodology in detail and provided a useful critique of how to improve on these pilot results, which add value to this paper.

There are some methods details that would benefit from further explanation: 

Lines 158-163: It would be helpful to show at least the equations used, so we know exactly how this was done and which fractional excretion values were used for each compound. Yes, this is probably available in the references cited, but these are critical calculations for the entire paper, and would be helpful to the reader to include here.

Most importantly, the authors need to explain and justify the use of geometric mean instead of arithmetic mean.  Presumably they did this because of outliers in the NHANES data. Is there a significant difference between the two arithmetic means?? (Fig 1)

Lines 283, 287:  Did these various studies use creatinine-corrected concentrations, or not?  Presumably, like has been compared with like, but it would be helpful to verify that.

Be sure to explain acronyms/abbreviations the first time they are used, even if they are standard terms in risk assessment literature.

It would be interesting to hear the authors thoughts on recruiting workers to participate - this must have been fairly difficult, if only 17 of the 26 participants in the educational effort agreed to this.

In general, well written and solid pilot study.

Author Response

Reviewer 1

This is a very solid study, well written and with a well thought out analysis.  If indeed there are only two other published research studies on phthalate exposures among nail salon workers in the US, it is a solid contribution to a much needed literature.  The authors explained their methodology in detail and provided a useful critique of how to improve on these pilot results, which add value to this paper.

We thank reviewer 1 for the positive overarching comments about our manuscript and have provided detailed responses to each comment regarding the method details in italics below (also attached in word).

There are some methods details that would benefit from further explanation: 

Lines 158-163: It would be helpful to show at least the equations used, so we know exactly how this was done and which fractional excretion values were used for each compound. Yes, this is probably available in the references cited, but these are critical calculations for the entire paper, and would be helpful to the reader to include here.

We agree and added the equation to estimate daily intake and the FUE values to the manuscript text as suggested, which now reads as follows (Lines 159–168):

“Daily intake for each phthalate was back-calculated from creatinine-corrected urinary metabolite concentrations (μg/g) using creatinine excretion rates and fractional excretion values of metabolites relative to parent compounds as follows: [(MEi x CE) / (FUE x 1000)] x (MWp / MWm), where i = individual phthalate, MEi = creatinine-adjusted urinary metabolite concentration for each phthalate (μg/g), CE = body weight-adjusted creatinine excretion rate (mg/kg/day), FUE = the fractional molar ratio of excreted metabolite to ingested parent phthalate, MWp = molecular weight of parent compound, and MWm = molecular weight of metabolite. Because MEHP was the only DEHP metabolite included in this analysis, an FUE value of 0.062 was used for DEHP estimation, while an FUE value of 0.69 was used to estimate DnBP, DiBP, and DEP [34].”

Most importantly, the authors need to explain and justify the use of geometric mean instead of arithmetic mean.  Presumably they did this because of outliers in the NHANES data. Is there a significant difference between the two arithmetic means?? (Fig 1)

Figure 1 shows the distribution of phthalate levels in both populations (nail salon workers and NHANES), which is clearly right-skewed (or log-normal) in both cases (i.e., the arithmetic mean is higher than the median or geometric mean because it is influenced by a smaller handful of values at the upper tail of the distribution, whereas if it was a normal distribution, they would be equal). The t-test used to compare the two distributions relies on the assumption of a normal distribution. Thus, we log-transformed the levels in the right-skewed distribution to better approximate a normal distribution, as is commonly done for log-normal chemical data in environmental and occupational exposure studies. In this case, the median or geometric mean (exponentiated mean on log scale) are considered to be more appropriate summary statistics for a t-test comparison than the arithmetic mean. We chose geometric mean rather than the median because we wanted to be consistent with NHANES.

Lines 283, 287:  Did these various studies use creatinine-corrected concentrations, or not?  Presumably, like has been compared with like, but it would be helpful to verify that.

Some studies reported creatinine-corrected concentrations while others corrected for urine dilution using an alternate adjustment method (e.g., specific gravity adjustment). Therefore, when possible, we compared levels to other studies using creatinine-corrected measurements (i.e., the previous nail salon biomonitoring study conducted by Hines et al. and the previous biomonitoring of the Vietnamese population conducted by Guo et al. in Table 5). Otherwise, we compared non-creatinine-corrected (wet-weight) concentrations if they were reported by the authors (i.e., the previous epidemiology studies included in Table 6). Note: studies that did not report either creatinine-corrected or non-corrected (wet-weight) concentrations were not included because they were not directly comparable (i.e., the previous nail salon biomonitoring study conducted by Kwapniewski et al.). Consequently, Lines 283–286 and Table 5 compare creatinine-corrected concentrations, while Lines 287–290 and Table 6 compare non-creatinine-corrected (wet-weight) concentrations.

Be sure to explain acronyms/abbreviations the first time they are used, even if they are standard terms in risk assessment literature.

Thank you for this comment. We have edited the manuscript to ensure that acronyms/abbreviations are spelled out the first time they are used.

It would be interesting to hear the authors thoughts on recruiting workers to participate - this must have been fairly difficult, if only 17 of the 26 participants in the educational effort agreed to this.

For this study, we recruited a subset of the larger educational study to collect urine samples from Vietnamese nail salon workers in the Bay Area. The educational study was designed by researchers partnering with community health workers and salon workers and owners through Asian Health Services and the California Healthy Nail Salon Collaborative. This collaborative coalition is dedicated to protecting worker health while maintaining the economic integrity of local businesses. This builds trust in the research process and paves the way for future work with this population. The larger educational study partnered with culturally competent members of the Vietnamese community who have forged relationships with nail salon workers and owners to obtain informed consent and collect air monitoring samples. However, because we leveraged the larger educational/air monitoring study once it was already underway, we were not able to obtain informed consent and collect urine samples for every participant of that study. But we think the pilot biomonitoring study demonstrates the feasibility of doing so in future studies among this population.

In general, well written and solid pilot study.

Thank you. We appreciate reviewer 1 taking the time to read our manuscript and provide feedback.

Reviewer 2 Report

Dear Authors,

the study although very simple in its design is informative and might be interesting to wider audience. However, measurement of phthalate metabolites in working and no-working days in the same persons would be more relevant.

The reliability of the analytical method is crucial for comparing exposure data. Has the laboratory ever participated in interlaboratory tests confirming the correctness of the obtained results?

Table 6. Please indicate in the table caption that the results refers only to female participants of your study.

Abbreviation "RPF" is used for the first time in the text, in line 188. It should be mentioned earlier in line 158 - where relative potency factor is described.

Author Response

Reviewer 2

We thank reviewer 2 for these thoughtful comments. We have provided detailed responses to each comment/question in italics below (also provided as a word attachment).

Dear Authors,

the study although very simple in its design is informative and might be interesting to wider audience. However, measurement of phthalate metabolites in working and no-working days in the same persons would be more relevant.

We agree although non-working days would be difficult to capture among this population of nail salon workers, who often work 6-7 days per week.

The reliability of the analytical method is crucial for comparing exposure data. Has the laboratory ever participated in interlaboratory tests confirming the correctness of the obtained results?

This is a great point. We conducted a round-robin with several labs, including the California Department of Public Health, to validate our measurement of phthalate metabolite concentrations in urine as we developed the method. Additionally, as noted in the Methods section of the manuscript, we adapted and validated the well-established Centers for Disease Control (CDC) laboratory protocol for the urinary analysis of phthalate metabolites to perform the analysis. The resulting Standard Operating Procedure has been published internally by the California Environmental Protection Agency Department of Toxic Substances and Control.

Table 6. Please indicate in the table caption that the results refers only to female participants of your study.

We added “among female nail salon workers” to the Table 6 title, which now reads, “Table 6. Comparison of non-creatinine-corrected phthalate metabolite concentrations (μg/L) measured among female nail salon workers in this study to previous epidemiology studies that reported significant associations between prenatal exposures and male developmental endpointsa

Abbreviation "RPF" is used for the first time in the text, in line 188. It should be mentioned earlier in line 158 - where relative potency factor is described.

We added the acronym, RPF, in parentheses after relative potency factors are first described in line 158, as suggested.
